# OpenReview forum: "How Reliable Is Human Feedback For Aligning Large Language Models?"
_ICLR.cc/2025/Conference — ICLR 2025 Conference Withdrawn Submission_

### Official Review · Reviewer_d1Yz · 2024-10-26

**Soundness:** 2
**Presentation:** 3
**Contribution:** 2
**Rating:** 5
**Confidence:** 4

**Summary:**

Much alignment research depend on using human feedback data such as Anthropic-HH but little attention has been paid to understanding the reliability of such feedback data and how they influence alignment outcomes. This paper seeks to understand the reliability of such data using a committee of gold reward models, which shows that 25% of dataset shows low or no agreement with the data. They further identify sources of unreliability stemming from mis-labelling, subjective preferences etc. Based on such insights, they prepare and release HH-clean which removes unreliable samples from HH-RLHF, which can be leveraged to train much better LLMs.

**Strengths:**

1.	The issue of unreliable human feedback is critical for model alignment as this essentially represents noise in training.
2.	The small-scale study to understand why humans disagree with HH-RLHF labels is interesting – with particular categories such as mis-labelling, subjective queries and different preference criteria being reasonable sources of difference.
3.	The proposal of source-aware cleaning is well-justified and empirically performs better than alternatives.

**Weaknesses:**

1.	The assumption that models trained on high-quality preference datasets such as UltraFeedback can identify errors in human feedback in HH-RLHF is a poorly supported one. [1] shows that on all categories on Rewardbench, the accuracy of the same model  (Llama2-7B) trained with HH-RLHF is comparable or better than that trained on UltraFeedback. Even in Table 4 in this paper, only 1 out of 8 gold RMs reach 76.8% on Chat-Hard while all other models score below 66% on the same category (as comparison, random chance is 50%). This suggests that that these RMs cannot serve as ‘gold’ RMs, unlike what the authors claim. Specifically, this is because UltraFeedback was annotated by GPT-4, which is known to have low variance but high bias (e.g. preferring longer responses, preferring responses from models of the same family/trained on GPT-4 output) [2]. Furthermore, all 8 RMs used as committee of gold RMs utilize GPT-4 generated data (to the best of my knowledge through datasets such as UltraFeedback or Nectar), making them less independent even as an ensemble. This means that they are not selecting for when human feedback is unreliable (i.e. disagreement between humans) but for when human feedback disagrees with GPT-4.
2.	The paper does not include enough details on how qualitative analysis by humans (line 236) is done. Specifically, it is missing the background of the annotators (e.g. demographics) as this is an important consideration in determining whether the preferences measured in this paper are indeed comparable with those from HH-RLHF. Furthermore, it’s not clear why the annotation interface departed from the one illustrated in the HH-RLHF paper if it's trying to understand differences in annotation on the same task. Particularly it provides more flexibility to annotators (i.e. less guidelines) and Likert-8 scale preference was simplified into binary choice (A is better or B is better) with the additional fields of both bad, both good and uncertain. One potential consequence of this flexibility is that inter-rater agreements seems to be on the low side (in Figure 4) ranging between 0.2-0.4 for Fleiss’ Kappa (fair to moderate). This raises concerns on whether the human annotation quality is high enough to reflect meaningful differences from HH-RLHF.

[1] https://arxiv.org/abs/2409.09603
[2] https://arxiv.org/abs/2305.14387

**Questions:**

NA

**Details Of Ethics Concerns:**

Not clear how human annotators were compensated for their participation in the annotations.

---

### Official Review · Reviewer_i2FP · 2024-11-01

**Soundness:** 2
**Presentation:** 3
**Contribution:** 2
**Rating:** 3
**Confidence:** 4

**Summary:**

1. This work conducted comprehensive research and in-depth analysis of human feedback data. Combined with a set of reward models, it was found that over 25% of the data showed low consistency, indicating a high degree of unreliability.
2. This work identified six key sources of unreliability through qualitative analysis: mis-labeling, subjective preferences, differing criteria and thresholds for helpfulness and harmlessness
3. This work proposed source-aware cleaning, an automated data-cleaning method that significantly improves data quality. Comparisons with 10 data-cleaning baselines demonstrated its superiority.

**Strengths:**

1. This paper studied the reliability of human feedback data, which holds certain significance for filtering human preference data.
2. This paper conducted further analysis through qualitative human analysis, which is missing in many other studies.
3. The experiments on data filtering strategies have practical significance.
4. This paper performed detailed experiments and conducted in-depth analysis.

**Weaknesses:**

1. The dataset and model are limited. The study conducted experiments using only one dataset and one model. Although detailed, we cannot confirm the generalizability of the experimental conclusions.
2. While the unreliability of preference data is a significant research area, the qualitative analysis in the paper focuses only on human-annotated data, raising my doubts about its applicability to synthetic data.
3. The use of multiple Golden RMs in data filtering further limits the scalability of the approach. The critical question is, if I have golden RMs, why not directly use them to optimize the model?
4. Although an in-depth analysis was conducted, the six sources of unreliability help little in constructing high-quality preference datasets.

**Questions:**

1. I believe adding at least one new dataset and model is necessary to improve this paper further.
2. The "both are bad" problem mentioned in the paper might be mitigated using methods like KTO.
3. Could the RM group be composed of weaker RMs instead?

---

### Official Review · Reviewer_qGoK · 2024-11-01

**Soundness:** 2
**Presentation:** 4
**Contribution:** 2
**Rating:** 5
**Confidence:** 3

**Summary:**

The paper addresses the issue of human feedback data quality in alignment datasets. Using a committee of reward models, it categorizes data points, complemented by annotations from multiple human annotators across an expanded set of labels. This approach enables a more thorough qualitative analysis, identifying six sources of unreliability stemming from human and data-related factors. Experiments show that cleaning the training data according to the proposed Source-Aware Cleaning (SAC) method leads to improved performance over baseline methods.

**Strengths:**

* The paper’s structure is clear, with contributions highlighted and linked to specific sections familiar to the reader.
* It includes numerous supportive plots and tables that effectively illustrate key findings.
* The qualitative analysis identifying six sources of unreliability provides valuable insights into annotation errors, an area often overlooked in alignment research.
* The proposed Source-Aware Cleaning (SAC) method is novel and based on a qualitative analysis, rather than pure heuristics.
* The authors make their cleaned dataset publicly available, offering a valuable resource for further research.

**Weaknesses:**

* The test set's size of 100 prompts may be insufficient for evaluating complex alignment tasks effectively.
* I’m uncertain about the role of human annotations in the SAC process. The method defines sources of unreliability using both RM categorization and human annotation information, but future datasets, especially large-scale ones, might not have access to human annotations. For the SAC method to be widely applicable, it would ideally rely on RM categorization alone. However, it seems that human annotations play a significant role, as indicated by the retention of sources 2, 3, and 4, which largely stem from human-driven insights and are spread across different RM categories. Could you clarify whether SAC can function effectively with RM categorization alone, or if human annotations are essential to its design?

**Questions:**

I have several questions to clarify specific points, and I’m open to revising my evaluation depending on your responses.

* The paper refers to the RMs as "gold," but I could not find any justification for this terminology, as these are models. I acknowledge that there is a correlation between RM categorizations and human annotations, but it’s not a perfect agreement. Could you clarify the rationale for labeling these RMs as "gold"?
* Could you clarify if eight RMs is sufficient for this categorization, or if using a different number (e.g., fewer or more RMs) might yield more reliable categorization results?
* For the human annotations described in Section 3, could you provide more detail on how the annotations were collected, including information about the annotators' backgrounds and selection criteria?
* There is a mismatch between the binary (comparative) labels used by the RMs and the more nuanced labels from human annotations (which include options like “both are X”). Was this intentional? If so, explicitly stating this rationale in Section 3 would be helpful.
* I assume that the test set includes examples from “source 2” (subjective queries), where any response could potentially be valid and are retained in the training set. Would it be more appropriate to suggest a filtered test set that excludes such queries for a clearer evaluation?
* Regarding the manual coding process mentioned (*“We manually code these data, as well as notes written by annotators, and identify six sources of unreliability”*), could you include additional details? Specifically, how were coding decisions made, how many individuals reviewed each data point, were there any specific guidelines, and how many data points were evaluated by these criteria?
* Were any of the RMs trained on Anthropic-HH?
* Could you explain the choice to conduct an evaluation using GPT-4o? How reliable are its evaluations, and were any human validations performed to ensure the accuracy of these assessments?

**Minor comments:**
* In Figure 4, consider renaming the x-axis to “quantile” or “fraction” or adjusting the scale to 0-100, as percentage values typically range between 0 and 100.
* Normalize $v_i$ by RM count to make its scale more intuitive.
* Overall, I found the mathematical definitions in Section 2 redundant, as they are not referenced later in the paper and could be more effectively conveyed in words.

---

### Official Review · Reviewer_MDKM · 2024-11-04

**Soundness:** 2
**Presentation:** 2
**Contribution:** 1
**Rating:** 3
**Confidence:** 4

**Summary:**

This paper analyzes the reliability of human feedback in the Anthropic-HH dataset and identifies six key sources of unreliability. The authors also propose the Source-Aware Cleaning technique for data cleaning. Experiments demonstrate that models trained on their cleaned version of the dataset outperform models trained on the original dataset.

**Strengths:**

1. This paper systematically analyzes the reliability of human feedback in the Anthropic-HH dataset.
2. The analysis of the dataset identifies six key sources of unreliability.
2. The authors also propose the Source-Aware Cleaning technique for data cleaning. Experiments demonstrate that models trained on their cleaned version of the dataset outperform models trained on the original dataset.

**Weaknesses:**

1. The contribution is quite narrow in scope. It seems the major contribution in this paper is just data cleaning.
2. The techniques applied for data cleaning have limited novelty, and it is not convincing whether this technique can be generalized to any other datasets.
3. Models trained on the cleaned version are only evaluated using GPT-4o. It makes the evaluation also questionable. How is it verified that GPT-4o is also aligned with humans?
4. The experiments are not rigorous and the authors should conduct more extensive experiments of the models trained on the cleaned version against the original dataset across more domains and evaluation datasets (e.g., more edge case scenarios).

**Questions:**

See above

---

### Note · Authors · 2024-11-27

**Comment:**

We thank all reviewers for their valuable feedback. After consideration, we decided to withdraw this submission and revise it for another round.

**Withdrawal Confirmation:**

I have read and agree with the venue's withdrawal policy on behalf of myself and my co-authors.